# Pathogenic and Prognostic Roles of Paraneoplastic Leukocytosis in Cervical Cancer: Can Genomic-Based Targeted Therapies Have a Role? A Literature Review and an Emblematic Case Report

**DOI:** 10.3390/diagnostics12081910

**Published:** 2022-08-07

**Authors:** Clelia Madeddu, Elisabetta Sanna, Sonia Nemolato, Olga Mulas, Sara Oppi, Mario Scartozzi, Giorgio La Nasa, Antonio Maccio

**Affiliations:** 1Department of Medical Sciences and Public Health, University of Cagliari, 09100 Cagliari, Italy; 2Department of Gynecologic Oncology, A. Businco Hospital, ARNAS G. Brotzu, 09100 Cagliari, Italy; 3Department of Pathology, ARNAS G. Brotzu, 09100 Cagliari, Italy; 4Hematology and Transplant Center, A. Businco Hospital, ARNAS G. Brotzu, 09100 Cagliari, Italy; 5Department of Surgical Sciences, University of Cagliari, 09100 Cagliari, Italy

**Keywords:** cervical cancer, leukocytosis, cytokine-secerning tumor, interleukin-6, myeloid-derived suppressive cells, prognosis, chemoresistance, alpesilib, ruxolitinib, cytoreductive surgery

## Abstract

Tumor-associated leukocytosis has been associated with poor prognosis in cervical cancer. Leukemoid reaction (i.e., white blood cell count > 40,000/μL) is defined paraneoplastic (PLR) when it occurs in the presence of a cytokine-secreting tumor (CST) without neoplastic bone marrow infiltration. Cervical cancers displaying PLR represent a peculiar entity characterized by a rapidly progressive behavior typically associated with chemo-radioresistance. The present paper aims to review the literature about the pathogenetic mechanisms of PLR and its prognostic role in cervical cancer. Moreover, it reports the emblematic case of a patient with an advanced cervical cancer associated with PLR that was chemotherapy resistant. The patient underwent a palliative cytoreductive surgery of high complexity, obtaining a temporary regression of PLR. The tumor sample stained positive for G-CSF and IL-6, thus indicating a CST. Notably, the tumor genomic analysis revealed a PI3CKA mutation. Therefore, at the instrumental evidence of a rapidly progressive disease relapse, which was accompanied by reappearance of PLR, we started a targeted treatment with a selective PIK3 inhibitor alpesilib combined with the JAK1-2 inhibitor ruxolitinib. We achieved a relief of symptoms and leukocytosis; however, severe side effects necessitated the treatment suspension. In conclusion, as therapeutic strategies for cancer with PLR are scarcely reported in literature, our study could contribute to expand our understanding of the topic and provide a basis for further research.

## 1. Introduction

Cervical cancer remains a significant worldwide health challenge, with a high mortality rate for those cases diagnosed at an advanced stage that manifest associated leukocytosis. Hence, identifying clinically significant prognostic markers is critical, especially in these patients who demonstrate a need for the highest complexity in therapy and management. In detail, the present paper aims to review the literature about the pathogenetic mechanisms of tumor-related leukocytosis and its prognostic role, focusing in particular on cervical cancers associated with paraneoplastic leukemoid reaction (PLR). Moreover, it reports an emblematic case of a patient with a chemotherapy-resistant advanced cervical cancer associated with PLR and characterized by a very aggressive clinical course. Then, we discuss the evidence and limitations about the current management of such a complex condition, the potential mechanism-based therapeutic strategies, and the basis for future research.

## 2. Cancer-Related Inflammation

The year 2021 marked a decade since Hanahan and Weinberg published their landmark paper on the hallmarks of cancer; it identified “tumor-promoting inflammation” as a key enabling characteristic [1]. In particular, the authors identified the inflammatory state of premalignant and malignant lesions—driven by cells of the immune system—as a strong promoter of tumor progression by different pathways. Rudolf Virchow, in 1863, was the first to link inflammation and cancer. He observed a high percentage of leukocytes infiltrating neoplastic tissue and hypothesized that the origin of malignancies could be consequent to chronic inflammation [2]. Since then, pathologists have increasingly reported that various tumors are heavily infiltrated by both innate and adaptive immune cells, mimicking inflammatory conditions occurring in non-neoplastic lesions [3]; we have recently demonstrated similar findings in a large and rapidly proliferating low abdominal lesion organized around the amniotic fluid that was composed predominantly of M1 macrophages, after a cesarean delivery [4].

In the last years, scientific investigations on the connections between inflammation and cancer pathogenesis have grown, leading to conclusive proof of the significant tumor-promoting effects exerted by immune cells, mainly of the innate immune system [5,6,7,8]. Inflammation can promote cancer progression by providing bioactive molecules to the tumor microenvironment (TME), inclusive of growth factors, survival factors, proangiogenic proteins, extracellular-matrix-modifying enzymes that enable angiogenesis, invasion, and metastasis (Figure 1). Moreover, during inflammation, innate immune cells can release, in addition to cytokines and chemokines, a large amount of reactive oxygen species as defense agents and products of hyperactivated energy metabolism (Figure 1); these act on DNA and are mutagenic and promote oncogenesis [6]. To date, inflammation and associated oxidative stress have also been known to concur with immune escape, with peculiar and reversible phenomena.

Typically, immune responses may represent an effort by the immune system to eliminate tumors. However, this overactivity of the immune system may allow the tumor to evade the immune response. This peculiar condition is considered a paradoxical mechanism for immune escape by the tumor using surprisingly specific cells of the immune system known for their immunosuppressive activity. Thus, inflammation is present in some tumors during the earliest stages favoring tumorigenesis, and thereafter accompanying cancer during its evolution, until cancer-associated systemic symptoms are determined [9].

The immune response against cancer can be divided into two phases. The initial phase is called “resistance”, where the body tries to get rid of cancer cells through the activation of the specific immune response. The tumor progression reflects the failure of the mechanisms of resistance as well as of the specific immunity; this is followed by a second phase of immune response, “the tolerance phase”, characterized by the prevalence of innate immunity consequent mainly to necrosis and the immunopathology phenomena. In this phase, the aspecific chronic inflammation sustained mainly by macrophages and neutrophils is prevalent. The persistence of these events negatively impacts immunosurveillance and determines severe systemic symptoms [10].

To better understand these phenomena, it is fundamental to highlight that in establishing tumor-associated inflammation, the phenomena of cellular **necrosis are important**. Necrosis related to cancer progression activates cells that release pro-inflammatory cytokines both in the TME and at the systemic level, as a physiological mechanism of repairing tissue damage, in the “tolerance phase” [10]. Consequently, cytokines can, in turn, engage innate immune cells, mainly macrophages and neutrophils, whose roles in their antineoplastic activity remain unclear [6,11,12]. Several studies have reported that the presence of aspecific immune cells in the TME, such as macrophages and neutrophils, can surprisingly promote tumor growth and orchestrate immune-suppressive status favoring tumor escape. Two types of tissue damage coexist (i) direct damage by the neoplasm and (ii) immunopathological damage. The latter, while seemingly beneficial in counteracting cancer growth, sustains the tumor escape and the cancer systemic symptoms through the associated production of dangerous chemokines and cytokines. As we have recently described, the cytokine storm during the evolution of cancer is the actual reaction responsible for these phenomena. Thus, most advanced cancers benefit from these phases of immune response that promote cancer progression instead of controlling its growth [10,13].

One of the most complex aspects to understand is the timing of the arrival and growth of specific immune cells within the neoplastic tissue that can affect the tumor immunophenotype itself [1]. A tumor is not only an assembly of neoplastic cells, but also a heterogeneous growth of different specialized cells, such as fibroblasts, mesenchymal cells, macrophages, mast cells, and neutrophils, as well as T and B lymphocytes, that finally surprisingly concur with the growth of cancer itself. This sets the problem of explaining the tumor as a host or as an integral part of an organism that, however, has lost its physiological biological destiny. Consistently, several studies have revealed an expanding record of signaling factors secreted by inflammatory cells, which exert tumor-promoting activities (Table 1). Conforming to the expression of these several mediators, tumor-infiltrating inflammatory cells have been reported to activate and aid in sustaining tumor angiogenesis, promote cancer cell proliferation, ease tissue invasion by being present in the peritumoral space, and favor the metastatic dissemination of cancer cells [14,15,16,17].

Hence, clarifying the meaning or the actual “scope” of the neoplastic process is difficult. Starting from the phase of oncogenesis, which can be induced by inflammation and associated redox stress, the neoplastic cell is considered as a pathogen. The neoplastic growth will depend on the genetic characteristics of this pathogen (cancer), modified by epigenetic phenomena, and is based on the integrity and effectiveness of the immune system [10]. In detail, during the resistance phase of the immune response using highly specialized cells (T lymphocytes, NK, and dendritic cells), the body tries to counteract neoplastic progression and metastasis. However, this phase fails too often both because the tumor does not express specific immunogenic antigens, and because the immune system is strongly deficient. Another reason for the failure of the resistance phase is the activation by the tumor of specific mechanisms that allow it to counteract the antineoplastic actions of the immune system, a phenomenon known as “immune escape” [18]. The latter phenomenon is extremely complex and surprisingly uses properties that are not specific to the neoplastic cell. These properties with “ad hoc” activities promote neoplastic growth and remodel the specific immunity by inhibiting it and subjecting it to mechanisms of refined suppression as the atypical activation of the immune checkpoint pathways. Where the resistance phase fails, alternative immune mechanisms are established, which aim to counteract neoplastic growth in a completely non-specific way. This, as written above, defines the phase of tolerance, mainly supported by the cells of innate immunity macrophages and neutrophils. This phase, by recognizing the defeat of the most refined defense systems, attempts to counteract the growth of neoplastic cells with aspecific mechanisms, which are, in fact, the same symptoms that accompany the cancer disease, such as anemia, anorexia, and weight loss with sarcopenia consequent to the remodulation of energy metabolism (hypercatabolism) [10]. Then, the phase of tolerance, albeit strategically designed to reduce tumor burden, unfortunately contributes to further neoplastic growth and the genesis of related symptoms [19]. Among all events, anemia stands out; anemia results from inflammation originating from a specific alteration of the iron metabolism, normally used to counteract—for example—the bacterial growth; however, in cancer, it contributes to the deficiency of the specific immunity by strongly interfering with the normal pathways of immune cell energy metabolism. We could speak in all respects of immunosuppression induced by the immune system, where specific inflammatory cytokines such as IL-6 and cells such as macrophages and neutrophils play the main roles. The clear association between the phenomena of chronic inflammation and the presence in the TME of non-specific immune cells that are also able to induce chemoresistance—which compromises the prognosis of tumors—is now highlighted [20,21,22]. However, the recruitment mechanisms of these cells by the neoplastic cells, which can characterize the histotype of some tumors, remain unknown.

More recently, besides fully differentiated immune cells localized in the tumor stroma, various partially differentiated myeloid progenitors have been recognized. These cells constitute intermediaries between circulating bone-marrow-derived cells and the differentiated immune cells usually observed in normal and inflamed tissues. Remarkably, such progenitors, such as their more differentiated descendants, exert proven tumor-promoting activity. Of notable interest, a category of tumor-infiltrating myeloid cells (identified by the co-expression of the macrophage marker CD11b and the neutrophil marker Gr1) suppresses cytotoxic T lymphocyte and natural killer (NK) cell activity, having been separately defined as MDSCs [23]. This feature advances the probability that recruiting certain myeloid cells may be in double measure advantageous for growing cancer, directly promoting tumor progression, in particular angiogenesis, while simultaneously offering a way to escape immune destruction.

Then, during the evolution of some neoplasms, a switch can develop in which cells of innate immunity prevail, i.e., neutrophils and macrophages, which in an autocrine and paracrine way recruit other cells at the level of the microenvironment and modulate their synthesis at the bone marrow level to determine pictures of hyperleukocytosis similar to leukemia; they are notoriously associated with poor prognosis, chemoresistance, and sickness with severe impairment of the general clinical state.

## 3. Pathogenesis of Paraneoplastic Leukocytosis

The notion of cancer-associated inflammation being a poor prognostic marker is evident. One of the most representative examples is cervical cancer associated with leukocytosis [24,25].

In many types of epithelial cancers, increasing evidence has effectively highlighted the implication of chronic neutrophilic and macrophagic inflammation in the pathogenesis of the tolerance phase, with consequent immunosuppression and cancer progression and metastasis. In this context, established tumors tend to promote myelopoiesis and sustain an increase in leukocytosis. Indeed, as explained above, tumor growth can overcome resistance mechanisms using the same immune system cells, indicating a change in the effectiveness of the specific immune response related to cancer hyper aggressiveness [16]. The most common evidence of these effects is the production in the TME of specific cytokines as a granulocyte colony-stimulating factor (G-CSF) and various interleukins, in particular IL-6 [26]. G-CSF is a glycoprotein that triggers the proliferation and maturation of marrow progenitor cells into totally differentiated and functionally activated neutrophils [27]. Commonly, G-CSF is produced by macrophages, monocytes, fibroblasts, and vascular endothelial cells. The ability of the tumor to secrete G-CSF may develop concurrently with tumor development (i.e., with the primary tumor cell generation), or it may be acquired subsequently through later dedifferentiation of the primary tumor. This capacity may also be achieved in metastatic sites even though the primary does not produce G-CSF [28]. Additionally, in tumors with multiple metastases, CSF-secreting capacity may be present in some metastatic sites and not in others [26].

Persistent leukocytosis (white blood cell (WBC) > 40,000/μL) in the absence of hematologic malignancy is defined as “leukemoid reaction” (LR). Occasionally, it can also be indicated as “extreme leukocytosis” [29]; in cases when the WBC count is >100,000/μL, it is also defined hyperleukocytosis [30]. The latter represents a medical emergency as it can cause increase in blood viscosity and tumor lysis syndrome [31]. Paraneoplastic LR (PLR) is defined as the LR that occurs owing to the presence of a non-hemato-lymphoid cytokine-secreting tumor (CST) and in the absence of neoplastic bone marrow infiltration [32]. The first PLR due to CST was observed in 1977 in a lung cancer patient [33]. Since then, it has been described in patients with malignant tumors of different origins [26], particularly of the cervix [34,35].

PLR in most cases occurs in the context of a CST, where, in addition to the most commonly reported G-CSF, other cytokines, such as interleukin (IL)-1a,b, IL-3, and IL-6, as well as tumor necrosis factor (TNF)-α, have been also described [26]. In this regard, animal studies have shown that IL-6 can support neutrophil/macrophage colonies in vivo and that, by acting on other immune cells, it can stimulate the synthesis of different CSFs via the bone marrow [36]. In cervical cancer, cases that produce G-CSF are very rare, while the production of IL-6 from different cell lines of uterine cervical cancers and cases of IL-6-producing cervical cancer have been reported [37]. Additionally, IL-6 is one of the main cytokines expressed by HPV16-associated cervical tumors [38]. Noteworthy, Stone et al., in a cervical cancer-bearing mouse model, showed that tumor cells, tumor-inflammatory infiltrate, and spleen myeloid APCs (CD19-MHC-II+) exhibit constitutive JAK2/STAT3 and STAT5 activation, which may be, at least partially, the cause of myeloid cell proliferation. Moreover, they discovered that tumor cells produced growth factors, IL-6, and myeloid cell-attracting chemokines; additionally, they found that in the TME, the majority of infiltrating cells were macrophages with a mixed inflammatory phenotype that expressed receptors to chemokines; myeloid-cell-attracting chemokines; and also receptors for IL-6, IL-10, IL-1, and TNF-α [39]. Moreover, other studies have reported that the supernatant from HPV-positive tumor cell lines, which contains the secreted IL-6 and prostaglandin E2, induces a suppressor phenotype in immune cells [40].

These mechanisms allow for autocrine stimulation of some CSTs’ growth. Moreover, as well as inducing bone marrow myelopoiesis, CSTs can also promote a “qualitative” effect by hindering myeloid cell differentiation in the peritumoral space [26]. Consequently, this results in the assembling of immature myeloid cells that are called myeloid-derived suppressor cells (MDSCs) [41,42].

Overall, the scientific literature indicate that (a) tumor-related leukocytosis may be determined by raised hematopoietic growth factor (e.g., G-CSF or IL-6) production by tumor itself or TMEs; (b) G-CSF release promotes myelopoiesis and the expansion of MDSCs, which constitute a subset of cells that augment with worsening leukocytosis; (c) MDSCs may be identified on routine CBCs as neutrophils; (d) MDSCs can hamper T-cell proliferation; (e) MDSCs can promote tumor progression and metastasis and, therefore, may account for the association between G-CSF, tumor-related leukocytosis, high NLR, and poor prognosis [43].

In cervical cancer, MDSCs have been involved in tumor progression by favoring tumor angiogenesis, the metastatic process, and immunosuppression [44]. Moreover, experiments in preclinical animal models have demonstrated that MDSCs are also related with higher resistance to radiotherapy [45]. Although chronic neutrophilic inflammation is involved in the initiation of several cancers, these cells are also implicated in the later phases of cancer development, i.e., progression and metastasis. In fact, the abovementioned mechanisms confirmed that cancers tend to foster myelopoiesis and to engage neutrophils to the TME, where these cells undergo reprogramming and transitioning MDSCs. In the TME, the MDSCs, through the synthesis of a variety of mediators, not only impair the anti-tumor action of tumor-infiltrating lymphocytes but also keep them out from the TME [42,46].

## 4. Leukocytosis and Prognosis in Cervical Cancer

Several authors have indicated a correlation between leukocytosis and poor prognosis in cervical cancer. Okazawa-Sakai et al. [47], in a population of 219 patients with FIGO stage IIB–IVA cervical cancer, demonstrated that pretreatment leukocytosis and pretreatment neutrophilia were significant independent predictors of distant relapse. Maulard et al. [48], in 238 patients with locally advanced cervical cancer, demonstrated that leukocytosis, together with anemia and thrombocytosis, was correlated with reduced disease-free survival and overall survival. Koulis et al. have reported that leukocytosis and increased NLR, together with anemia, were associated with worsened PFS and OS in 257 patients with cervical cancer treated with radical chemoradiotherapy [49]. Similarly, Garcia-Arias et al., in a retrospective analysis of 294 consecutive patients with new diagnosis of untreated locally advanced cervical cancer, demonstrated that leukocytosis and hemoglobin level were predictive of poor outcome [50]. Mabuchi et al. [51] revealed in a retrospective analysis of 536 patients with cervical cancer that leukocytosis at the time of recurrence was an independent prognostic factor and correlated with short survival after recurrence (median: 9 months).

Others have reported on more general hematologic markers related to inflammation such as neutrophilia [52] and elevated neutrophil-to-lymphocyte ratio (NLR) [53]. As the inflammation process is linked with cancer growth, different circulating inflammatory markers have been evaluated for their potential role as prognostic and predictive factors of clinical outcomes in different cancer types. The NLR is calculated from the ratio between the neutrophil and lymphocyte counts obtained from a full blood count. It has been suggested as one of the most helpful prognostic indices in cancer [54]. In patients with cervical cancer, the NLR has been reported as a poor prognostic factor in addition to tumor stage. Some clinical studies have reported the correlation between pre-treatment NLR and the prognosis of cervical cancer [49,55]. Meanwhile, more recently, we have demonstrated in a population of patients with ovarian cancer that the pretreatment NLR value does not constitute a prognostic and predictive factor, but only the evidence of the extent of the systemic inflammation in progress; conversely, the increase in the number of lymphocytes during chemotherapy treatment, and therefore the reduction of NLR, constitutes one of the most valid predictors of response to treatment [56].

## 5. Case Report

Here, we report an emblematic case of a patient with PLR associated with advanced cervical cancer with a very aggressive clinical course, where, in addition to a maximal cytoreductive surgery of high complexity, we tested some drugs that selectively inhibit the cytokines involved in the pathogenesis of leukocytosis and the onset of related symptoms. A 43-year-old woman was referred to our unit from another institute, with a diagnosis of chemoresistant stage IVA squamous cervical cancer. Her clinical history started in July 2020 when she experienced vaginal discharge and bleeding. For these symptoms, she was referred in April 2021 to the Department of Obstetrics and Gynecology at another institute where she underwent a bimanual examination and transvaginal ultrasound that revealed a solid lesion occupying the cervix measuring 45 × 26 × 53 mm and extending to parametria and cranially to the isthmus with likely involvement of the uterine corpus and suspected bladder infiltration. The lesion showed a central and peripheral vascularization color score of 4, indicating a high vascular flow; resistance index was 0.3. The analysis of serum tumor markers indicated an elevated squamous cell carcinoma antigen level of 17.8 ng/mL (normal range < 2.0), whereas the levels of carcinoembryonic antigen (<0.5 ng/mL), carbohydrate antigen (CA)19.9 (14.8 U/mL), and CA 125 (7.8 U/mL) were within the normal range.

Subsequently, a computed tomography (CT) scan was performed that better defined a neoplastic lesion at the level of the cervix–uterine body measuring 64 × 55 mm. This lesion was without a cleavage plane posterior to the rectum and anterior to the bladder; was associated with thickening of the mesorectal fascia and presacral adipose layers; and had multiple adenopathies that existed bilaterally along the external iliac and in the obturator axis, the largest with colliquative aspects measuring 32 × 27 × 40 mm in the left external iliac. Then, a biopsy of the cervical lesion was performed, which indicated an invasive squamous cervical carcinoma, grading 2, p16 positive. The patient also underwent cystoscopy with biopsy, which confirmed bladder involvement. The clinical stage according to the FIGO was IVA. Considering the poor clinical condition and the severe pain and disease-related symptoms, the patient at that center was immediately submitted to systemic chemotherapy with carboplatin + paclitaxel. However, after the first cycle of chemotherapy, her clinical conditions worsened with uncontrolled pain and high fever.

Consequently, she was admitted to the inpatient ward on 22 June 2021. The laboratory analysis suggested leukocytosis, with the results as follows: WBC: 22,970/µL; neutrophils 21,300/µL; lymphocytes 600/µL; NLR 35.5; monocytes 400/µL; platelets 455,000/µL; and hemoglobin (Hb) 8.1 g/dL. Levels of fibrinogen, C-reactive protein (CRP), and procalcitonin were 455 mg/dL, 152.7 mg/L, and negative (0.59 ng/mL), respectively. Because of the high fever (range: 39–40.5 °C) and leukocytosis, samples of her blood, urine, and vaginal swab were subjected to bacterial culture to establish the possible source of infection; however, all culture results were negative. No evidence of HIV and hepatitis infections by serological analyses were noted. She received empiric antibiotic therapy with ceftriaxone and quinolone antibiotics; however, her fever and leukocytosis persisted. A repeat CT (16 June 2021) was performed that indicated a significant volumetric increase in the colliquated lymph node package in the left external iliac measuring 75 × 74 × 79 mm (versus 32 × 27 × 43 mm). This neoformation reached the left bladder wall medially that was extensively infiltrated, engulfed the left ureter, and was laterally in close contact with the iliopsoas muscle without a cleavage plane.

On July 1, 2021, her laboratory results were as follows: WBC, 48,510/µL; neutrophils, 42,300/µL; lymphocytes, 1500/µL; NLR, 28.2; monocytes, 2400/µL; platelets, 433,000/µL; Hb, 7.9 g/dL; and CRP level, 244.1 mg/L. Thus, as chemotherapy was not deemed to be feasible, the patient asked to be transferred to our Department of Gynecologic Oncology at the Regional Referral Center for Cancer Disease, ARNAS G.Brotzu, Cagliari. Upon admission to our department (7 July 2021), the laboratory parameters were as follows: WBC, 60,850/µL; neutrophils, 58,840/µL; lymphocytes, 430/µL; and NLR, 136. In view of the laboratory tests, the diagnosis of PLR was determined. Serum IL-6 level was markedly elevated at 320.3 pg/mL (normal range 0–7 pg/mL), and ferritin level was 2500 µg/L. Bone marrow aspiration and biopsy to exclude hematological malignancy were performed. No abnormalities were identified.

The patient received supportive treatment with steroids. After 10 days, the fever subsided, and the patient had a mild clinical benefit; hence, we performed a second cycle with weekly fractionated platinum-taxane chemotherapy, during which we observed a further progression of the disease on CT. We observed the neoplastic lesion infiltrating the bladder; necrotic colliquative adenopathies in the external and internal left iliac; voluminous secondary formation in the context of the left iliopsoas muscle with areas necrotic in the context (108 × 85 × 80 mm); and severe bilateral hydroureteronephrosis with the left ureter, which appeared incorporated in the mass in its distal tract (Appendix A). Then, in accordance with the patient, we opted for cytoreductive palliative surgery. On 20 July 2021, the patient underwent radical hysterectomy with bilateral salpingo-oophorectomy; total vaginectomy; partial cystectomy; omentectomy; peritonectomy; pelvic lymphadenectomy; and laterally extended resection of the pelvic sidewall with en bloc excision of the left pelvic mass including external and internal iliac vein, partial resection of the psoas muscle, obturator nerve, obturator internus muscles, and partial ureteral resection with reimplant of the same into the residual bladder.

Histological examination (Appendix A) confirmed a squamous cell cervical carcinoma, grading 3, with necrosis of the cervix, involving all the cervical canal up to the exocervix, which was extensively ulcerated, up to the internal uterine orifice, infiltrating the vagina for the upper 2/3, and the paracervical tissue; the involvement of peritoneum, right parametrium, bladder and ureteral wall, and all left pelvic lymph-node by carcinoma was observed. Additionally, the immunohistochemical (HIC) examination showed an intratumoral lymphomonocytic and granulocytic infiltrate (Appendix A) consisting of the following cells: frequent T lymphocytes CD8+, CD7+, and PD1+; numerous macrophages CD68+, PGM1+, and KP1+; rare neutrophils CD15, MPO, CD33 positive; and scattered eosinophils (associated to necrotic areas). Tumor-associated lymphocytes predominantly were composed of T cells (70%) CD3+, CD4+, CD5+, and PD1+, and B cells (approximately 30%) CD20+, PAX5+, and CD79a+; moreover, rare neutrophils and scattered macrophages CD68+ (PGM1+ and KP1+) were observed. The levels of G-CSF and IL-6 were analyzed by HIC assay, which demonstrated IL-6 and G-CSF intratumoral and peritumoral expression both in cancer cells and in macrophages (Appendix A). The tumor sample was also analyzed for genomic mutations, in which the mutation p.542/545 (E542K-E545K/Q) in exon 9 of the PIK3CA gene was identified.

Postoperatively, the laboratory parameters indicated a significant improvement of the leukocytosis as follows: WBC, 6990/µL; neutrophils, 5170/µL; lymphocytes, 1440/µL; and NLR, 3.6. On 11 August 2021, a CT scan of the total body was performed, and no indications of residual disease were observed. The uroCT revealed good outcomes of the ureteral implant. However, in September 2021, she presented with a high fever and was referred to us again. The laboratory examinations indicated the following results: WBC, 31,650/µL; neutrophils, 27,200/µL; lymphocytes 1500/µL; NLR, 18.1; monocytes, 2100/µL; platelets, 475,000/µL; Hb level, 8.0 g/dl; CRP level, 298.1 mg/L; ferritin level, >1650 ng/mL; and IL-6, 445.6 pg/mL. The CT indicated liver metastasis measuring 21 mm, and the appearance of a solid mass measuring 8.5 × 4.5 cm extended from the left presacral region cranially to the L4 infiltrating the iliopsoas muscle and caudally in the iliac fossa infiltrating the iliac muscle and inglobating the external iliac artery. Multiple metastatic nodal masses were observed in the right iliac (3 cm) and lombo-aortic region (1.8 cm). The progression of the disease was associated with the immediate increase in neutrophils and with symptoms associated with the inflammatory state such as fever and elevated IL-6, CRP, and ferritin levels.

Therefore, on the basis of these laboratory parameters indicative of severe inflammation and the tumor mutational profile positive for PIK3CA gene mutation, in October, we decided to initiate therapy with the selective inhibitor Janus-Associated Kinases (JAKs) JAK1 and JAK2, ruxolitinib (JAKAVI©), at 10 mg/day, and the oral selective inhibitor of the phosphoinositol-3-kinase (PIK3), alpesilib (PIQRAY©), at 300 mg/day. One week after the treatment initiation, the patient was referred for the remission of fever, disappearance of pain, and clinical benefit. The blood count revealed a slight reduction of leukocytosis with WBC, 20,670/µL; neutrophils, 18,600/µL; and platelets, 319,000/µL. After 2 months, cutaneous rash, edema, and hyperglycemia requiring insulin therapy occurred, which led to the disruption of both the JAKAVI and PIQRAY therapies; the drug interruption was associated with the re-onset of fever (38.8 °C), and the patient was readmitted to the hospital. In the CT, disease progression with a tumor mass with extremely aggressive growth was observed. In December 2021, the patient died of multiple organ failure, approximately 5 months postoperatively.

## 6. Discussion

Cervical cancers displaying PLR represent a distinct peculiar entity and have a rapidly progressive nature, which is also related to their ability to produce G-CSF. This favors MDSC expansion and creates a tumor-promoting TME, thus inducing tumor growth, associated with a systemic increase in leukocytes and inflammatory status [25]. No less important is the associated chemo-radio-resistance, typical of most of these tumors [45].

Here, we report the case of a patient with an advanced squamous cervical cancer associated with PLR and that was chemotherapy resistant who underwent palliative cytoreductive surgery; the patient’s cervical tumor sample showed positive staining for G-CSF and IL-6, thus indicating a CST. Moreover, the genomic analysis of the tumor demonstrated the presence of a PI3CKA mutation. These data led us to implement a targeted therapeutic approach with alpesilib (selective PIK3 inhibitor) and ruxolitinib (JAK1-2 inhibitor). Indeed, ruxolitinib, a drug currently approved for intermediate or high-risk myelofibrosis and polycythemia vera with severe leukocytosis that has failed hydroxyurea, targets the Janus kinase signal transducer and activator of transcription (JAK-STAT) signaling, thus inhibiting both the downstream pathway that mediates the neutrophil-proliferative effects of CSF [57] and the upstream pathway that induces the STAT-3-mediated IL-6 synthesis [58]. The use of the latter drug is noteworthy, which, to our knowledge, has not been previously described in the literature for the treatment of these particular tumors.

To date, some case reports of cervical carcinoma with leukocytosis as a paraneoplastic syndrome have been described, and most of them have suggested an extremely fast progression of these cancers with a very poor patient prognosis (Table 2). In detail, Kio et al. [34] reported a case of cervical cancer with severe leukocytosis, which presented an extremely aggressive clinical course associated with the production of G-CSF and IL-6 and increased expression of their receptors by tumor cells; these may have contributed to PLR and autocrine stimulation of tumor growth. Matsumoto et al. [59] reported four patients with G-CSF-producing cervical cancers who experienced early recurrences with short overall survival (<9 months) despite initial chemo-radiotherapy and aggressive surgical treatments. This report heavily suggests the aggressive character of the G-CSF-producing cervical cancer. Mabuchi et al. [60] also reported their experience with two cases of G-CSF-producing cervical adenocarcinomas, which exhibited a very aggressive clinical course with a median survival of 9 months after conventional primary treatment. Yabuta et al. reported two other cases of G-CSF-producing squamous cell carcinoma of the cervix, where marked leukocytosis was associated with extremely aggressive tumor growth [61]. Ahn et al. [62] reported a case of invasive squamous cell cervical carcinoma with leukocytosis (up to 69,000/µL) that normalized after chemo-radiotherapy without evaluating the GCSF levels of the tumor. Similarly, Nasu et al. [63] reported a case of locally advanced cervical cancer (stage IIIB) with leukocytosis (WBC 30,400/µL) associated with GM-CSF production by tumor cells that resolved after successful treatment with chemoradiotherapy. Nimieri et al. [64] reported a case of a very aggressive metastatic cervical cancer associated with PLR (WBC 93,000/µL) and treated with chemotherapy and local radiotherapy, who died 6 weeks after the diagnosis. Qing Li et al. [35] reported a case with a relatively early stage (FIGO IIA1) of cervical squamous carcinoma, which presented disease recurrence associated with LR 2 months after laparoscopic radical hysterectomy; in this case, administration of a combined chemotherapy regimen obtained an impressive tumor response with resolution of paraneoplastic LR.

On the basis of the data from the literature, the concept of chemoresistance, which heavily influences the prognosis of these types of tumors, may be valid but not in an absolute and univocal way. Therefore, the role of paraneoplastic leukocytosis should be better defined and studied in those cases of cervical cancer that indicate a fast progression with chemo-radioresistance, highlighting that leukocytosis is not sufficient to establish “a priori” chemo-radioresistance, and therefore, “a priori” poor prognosis. Instead, it seems crucial to demonstrate the production of G-CSF and/or IL-6 by these tumors. However, it should be noted that, whereas in some of the above-reported cases the authors use the term PLR for WBC values of approximately 30,000/μL, the definition of PLR as per literature is deemed to be a WBC count of above 40,000/μL [32,65]. Hence, leukocytosis with high NLR alone related to the progression of the neoplastic disease may not be a determining prognostic factor, as we have recently clearly demonstrated in ovarian cancer. Indeed, in advanced ovarian cancer patients, we found that pretreatment NLR value does not represent a prognostic, while the increase in the number of lymphocytes, and consequently the reduction of NLR, during the chemotherapy regimen constitutes one of the most valid predictors of response to treatment and patient prognosis [56].

In this regard, discussing the theme of inflammation and related symptoms, in cachectic patients at the time of cancer diagnosis, we have recently highlighted how the clinical course of cachexia, despite being function of the extent of disease, regresses according to the sensitivity to the selected treatments [10]. This certifies that strongly cachectic and heavily inflamed patients with ovarian cancer manifest the total resolution of the clinical picture referable to inflammation, after appropriate effective surgical and chemotherapeutic treatments; this included weight and muscle mass regain, the disappearance of anemia and thrombocytosis, and increased number of lymphocytes and decreased NLR ratio. To date, this evidence has not been demonstrated in patients with cervical cancer with leukocytosis and associated tumor production of G-CSF, suggesting that the crucial challenge may not be inflammation, but more likely the specific mechanisms of resistance to chemo-radiotherapy of these cancer cells that secrete factors inducing leukocytosis. Thus, the problem of treating paraneoplastic production of cytokines inducing hyperleukocytosis as a preparatory, mandatory criterion for modifying the response to treatment remains the most interesting aspect.

Treatment strategies for cancer with PLR are few and scarcely reported. Notably, in the literature, only a few patients who responded to the treatment recorded a longer survival [29]. Surgical resection, radiotherapy, and chemotherapy have demonstrated efficacy in reducing WBC counts only for responsive tumors [66,67,68]. However, typically, the malignant cervical tumors with LR respond poorly to chemotherapy, and the patients die shortly after, as in the present case reported. From this, it would appear that treating the underlying malignancy be the main treatment for PLR. This statement would be consistent with the hypothesis that the evolution of the neoplasm induces leukocytosis; consequently, leukocytosis is contextual, and, in turn, because of the production of specific neutrophilic/granulocyte cytokines, it is capable of inducing chemo-radioresistance and immunosuppression [45]. In fact, the anti-apoptotic activity, stimulation of tumor angiogenesis and inhibition of host immunity exert a fundamental role in the aggressive behaviors of these tumors [25].

In this case, the patient, suffering from cervical cancer with associated leukocytosis, demonstrated chemoresistance associated with a rapid progression of the disease. It was only after an extensive surgical intervention that the leukocytosis momentarily regressed but later reappeared with the radiological evidence of the disease relapse. This would support the hypothesis that leukocytosis should not be considered a phenomenon of the immune system associated with neoplastic growth, but rather as a peculiar feature induced by specific cytokines produced by the tumor.

In the literature, 76% of patients who had PLR, if associated with G-CSF production, died within 6 months of presentation [29,65]. Hence, the above hypothesis logically explains the aggressive behavior of these tumors and their poor prognosis. Starting from the evidence that supports the pro-tumorigenic actions of MDSCs of neutrophilic origin and their association with worsening leukocytosis and prognosis, a variety of adjunctive therapeutic approaches targeting the recruitment of these cells and/or the deleterious effects of their mediators have been developed [69]. Most of these are in the pre-clinical or very early clinical phases of assessment. However, significant exceptions are some pharmacologic, allosteric inhibitors of neutrophil/MDSC CXCR1/2 receptors. These drugs moved into the late-stage clinical evaluation as complement to either chemotherapy or immune checkpoint inhibitors or other targeted therapies in patients with different types of advanced malignant cancers [46].

Consistently, in the complex cases of the cervical tumor with associated PRL and chemoresistance, the design of a potentially effective treatment protocol could benefit from incorporating drugs that target the different biochemical pathways implicated in promoting the growth of a very aggressive tumor and the associated severe leukocytosis and inflammation. On the basis of these observations, in the present case—where we identified a G-CSF- and IL-6-secreting tumor associated with increased circulating levels of IL-6, PCR, and ferritin, as well as a somatic tumor mutation of the PI3KCA gene, which is present in at least 40% of cervical cancers [70]—we attempted to co-target both the specific mutation driving the tumor growth and the pathways involved in the production of G-CSF and IL-6 by using a monoclonal antibody against PI3KA [71] and a JAKs inhibitor [58,72]; our aim was that such an approach would result in effective treatment. Of note, the presence of the PI3KCA mutation was itself associated with cervical tumors characterized by extremely aggressive behavior, chemoresistance, and poor prognosis regardless of leukocytosis. Noteworthy, a very recent paper demonstrated that PIK3CA expressing cervical cancers demonstrated an increase in the MDSC and M2 inhibitory myeloid populations. The myeloid microenvironment, characterized by Ccr2hi myeloid cells of PIK3CA mutated tumors synthetizes Ccl2 and inhibits CD8+ T-cell proliferation. Activation of PI3K signaling in cancer cells generates an immunosuppressive TME denoted by the increased presence of inhibitory CD45+ CD11b+ Ccr2hi myeloid cells [73].

The use of the specific anti-PI3KCA and anti-JAK drugs in our case only achieved a momentary reduction in symptoms and leukocytosis; however, side effects such as hyperglycemia and skin rash necessitated the suspension of the treatment, which then coincided with rapid clinical relapse of symptoms and a negative disease outcome. On the basis of the aforementioned concepts, we believe that the use of these drugs, if used in the very early stages of the disease and with careful identification of the most appropriate dosages, may prove beneficial.

## 7. Conclusions

Of note, our study represents an emblematic example and unique attempt that tested a mechanism-based targeted approach in such complex condition. More studies should be conducted to explore the phenomena that induce chemoresistance, rather than phenomena related to inflammation, in order to understand whether the same mediators of leukocytosis are the inducers of the peculiar chemo- and radioresistance of these tumors. Furthermore, future studies should highlight the reason why chemotherapy for these tumors does not exert either a myelosuppressive action or an effective antineoplastic action. In conclusion, as treatment strategies for cancer with PLR are few and rarely reported; our study represents an emblematic example and unique attempt that tested a mechanism-based targeted approach in such complex condition. More studies should be conducted to explore the phenomena that induce chemoresistance, rather than phenomena related to inflammation, in order to understand whether the same mediators of leukocytosis are the inducers of the peculiar chemo- and radioresistance of these tumors.

## Figures and Tables

**Figure 1 diagnostics-12-01910-f001:**
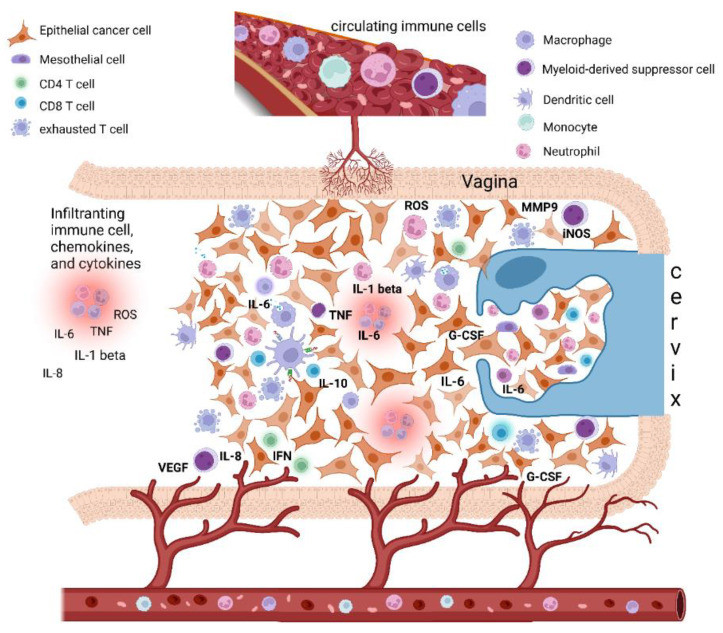
Epithelial cervical cancer microenvironment, tumor-associated inflammation, and paraneoplastic leukocytosis. The cervical cancer tumor microenvironment (TME) is composed of different cells, including epithelial cancer cells, fibroblasts, mesenchymal cells, and immune cells, i.e., macrophages, mast cells, dendritic cells, and neutrophils, as well as T and B lymphocytes. Cancer-related inflammation can promote cancer progression by providing bioactive molecules to the TME, inclusive of growth factors, survival factors, proangiogenic proteins (VEGF), extracellular-matrix-degrading enzymes (MMP-9) that enable angiogenesis, invasion, and metastasis. These signaling molecules are released by the immune inflammatory cells as well as by cancer cells themselves. Moreover, during inflammation, immune cells can release several cytokines and chemokines (IL-6, IL-1beta, TNF-alpha, IL-8, INF-gamma) as well as a large amount of ROS as defense agents and products of hyperactivated energy metabolism. In turn, cytokines engage innate immune cells, mainly macrophages and neutrophils, that finally promote tumor growth and immune-suppressive status favoring tumor escape. In this context, cervical cancer tumors may also tend to promote myelopoiesis and sustain an increase in leukocytosis. This event seems to be mediated by the production in the TME of specific cytokines as a G-CSF and IL-6, both by cancer cells and TME inflammatory infiltrate. Beside inducing myelopoiesis, these factors can also promote the assembling in TME of immature myeloid progenitors known as myeloid-derived suppressor cells (MDSCs), which are able to directly promote tumor progression, in particular angiogenesis, and also suppress cytotoxic T lymphocyte and natural killer (NK) cell activity, thus contributing to immune escape. Abbreviations: IL, interleukin; TNF, tumor necrosis factor; IFN, interferon; ROS, reactive oxygen species; iNOS, inducible nitric oxide synthase; MMP, metalloproteinase; G-CSF, granulocyte-colony stimulating factor; VEGF, vascular endothelial growth factor. Created with BioRender.com.

**Table 1 diagnostics-12-01910-t001:** Definition of the main immune cells involved in cancer-related inflammation and paraneoplastic leukocytosis and their respective functions.

Category	Cells	Main Functions	Cytokines/Effectors
**Myeloid**	Neutrophils	Phagocytic cells that rapidly migrate to site of cancer/inflammation and recruit other immune cells Direct cytotoxicity Regulation of cytotoxic T lymphocytes response	Proinflammatory cytokines (IL-6, IL-1b), ROS
	Tumor- associated macrophages	Antigen-presentation and T cell activation in the first phase of antitumor immunity; Tumor-promoting activity with inhibition of T cell activity and proangiogenetic activity	M1: IL-6,TNF-a, IL-1b, IL-6, IL-12, IL-23, iNOS, COX-2; M2: IL-10, VEGF, Arginase, MMP9, IL-8
	Dendritic cells	Antigen-presenting cells, that display antigen to activated T lymphocytes	PDL-1 (immature dendritic cells)
	Myeloid derived suppressor cells	Suppression of T cells and NK cells activity; Tumor promoting activity; Proangiogenetic activity	ROS iNOS MMP9 Arginase
**Lymphoid**	T lymphocytes		
	Cytotoxic T cells (CD8+)	Direct lysis of cancer cells; production of cytotoxic cytokines	
	T helper (CD4+)	Help cytotoxic T lymphocytes (CTLs) in tumor rejection; B cell activation; production of cytokines	INF-γ
	Treg cells (CD4+)	Inhibition of CD8+ CTLs	
	B cells	Production of tumor-specific antibodies Activation of mast-cells	Tumor-specific antibodies
	NK cells	Direct cytotoxicity of cancer cells Production of cytotoxic cytokines	INF-γ, VEGF

Abbreviations: ROS, reactive oxygen species; IL, Interleukin; TNF, Tumor necrosis factor; iNOS, inducible nitric oxide synthase; COX, cyclooxygenase; VEGF, vascular endothelial growth factor; MMP, metalloproteinase; INF, interferon; NK, natural killer.

**Table 2 diagnostics-12-01910-t002:** List of similar cases or study population on paraneoplastic leukocytosis in cervical cancer available in literature.

Ref.	No. Cases	Stage	Pathology	Primary Treatment	Time to Recurrence	Recurrence Site	WBC (cells/microL)	CST	Treatment of Recurrence	Survival
Kio [34]	1	IB	SCC	Radical hysterectomy, bilateral salpingo-oophorectomy + pelvic lymphadenectomy	30 days	Pelvis	45,000	Yes	None	68 days, DOD
Matsumoto [59]	4	(a) IB2	SCC	CCRT	6 months	Uterus, lung	25,670	Yes	Surgery of metastasis and chemotherapy	15 months, DOD
		(b) IIB	SCC	CCRT	3 months	Liver	34,470	Yes	Chemotherapy	9 months, DOD
		(c) IVB	SCC	RT followed by chemotherapy	1 month	Lung, nodes	25,270	Yes	None	3 months, DOD
		(d) IB2	SCC	Surgery + adjuvant RT	7 days	Brain, lung	13,960	Yes	Chemotherapy	5 months, DOD
Mabuchi [60]	2	(a) IIA	ADC	RT	30 days	Liver, lung, supraclavicular lymphnode	11,830	Yes	None	3 months, DOD
		(b) IB2	ADC	Radical hysterectomy, bilateral salpingo- oophorectomy, pelvic lymphadenectomy followed by adjuvant RT	28 days	Pelvis, lung, supraclavicular, and paraaortic lymphnode	15,580	Yes	Chemotherapy	6 months, DOD
Yabuta [61]	2	(a) IIB	SCC	Radical hysterectomy + pelvic lymph node dissection followed by CCRT	30 days	Pelvis	52,670	Yes	NR	12 months, DOD
		(b) IIB	SCC	CCRT	NR	Pelvis	41,030	Yes	NR	2 years, DOD
Ahn [62]	1	IIB	SCC	Neadjuvant chemotherapy	6 weeks	Cervical	69,000	Yes	CCRT + brachytherapy *	4 months, DOD
Nasu [63]	1	IIIB	SCC	RT	-	-	30,400	Yes	-	8 months, alive
Qing [35]	1	IIA1	SCC	Laparoscopic radical hysterectomy, pelvic lymphadenectomy, vaginoplastic, and ovarian transposition	68 days	Vaginal	70,000	NA	Chemotherapy + RT	20 months, alive
Nimieri [64]	1	IVB	SCC	Chemotherapy and radiotherapy	-	-	93,000	NA	-	6 weeks, DOD

* The patient had a further metastatic lung recurrence after one month, that was treated with platinum-based chemotherapy; Abbreviations: SCC, squamous cell carcinoma; ADC, adenocarcinoma; CCRT, concurrent chemoradiotherapy; RT, radiotherapy; NA, not assessed; NR, not reported; DOD; dead of disease.

## Data Availability

Original clinical, laboratory, and instrumental data can be found in the patient chart archived at the Department of Obstetrics and Gynecology and are available on request from the corresponding author.

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
