# Peer review of "Pathogenic and Prognostic Roles of Paraneoplastic Leukocytosis in Cervical Cancer: Can Genomic-Based Targeted Therapies Have a Role? A Literature Review and an Emblematic Case Report"

_diagnostics, 2022, doi:10.3390/diagnostics12081910_

Round 1
Reviewer 1 Report
Diverse aspects of cancer microenvironment and immune reaction was mentioned. But it is hard to understand whole picture of this study.
Case report CT scan and pathology pictures does not improve quality of this review study.
It is more desirable by making a table of combining similar cases or study population.
Also, I suggest making a flowchart or illustration to make readers understand easily about molecules mentioned in manuscripts. And another table to define immunologic cells and their functions is needed.
Other Comments:
Topic is original & relevant in the field. It addressed enough gap in manuscript.
Review article is different from case report. It’s a little confusing to explain in detail about specific case. I suggest author focus more on systemic articles classification and related table and illustration if possible.
Appropriate references are mentioned.
Author’s conclusion is “treatment strategies for cancer with PLR are few and rarely reported, our study could significantly contribute to the literature by expanding our understanding of the topic and providing a basis for further research” is a little different in “. our study represents an emblematic example and unique attempt that tested a mechanism-based targeted approach in such complex condition. More studies should be conducted to explore the phenomena that induce chemoresistance, rather than phenomena related to inflammation, to understand whether the same mediators of leukocytosis are the inducers of the peculiar chemo- and radio-resistance of these tumors”. Author should explain objective of this “Review” article more clearly.
Author Response
Point-by-point reply to Reviewers’ comments
Reviewer 1
Comments and Suggestions for Authors
Diverse aspects of cancer microenvironment and immune reaction was mentioned. But it is hard to understand whole picture of this study.
Reply: I have added a part at the end of the first paragraph to better clarify the aim and the whole picture of the study. Moreover, I have added a new heading “Cancer-related inflammation” for the first part focused on the description of the mechanisms of cancer-related inflammation (lines 43-51 of the revised version)
Case report CT scan and pathology pictures does not improve quality of this review study.
Reply: I have removed the CT scan and pathology picture from the main manuscript. If you agree I have maintained them as supplementary files.
It is more desirable by making a table of combining similar cases or study population.
Reply: I have added a table illustrating the similar cases or study population available in literature on paraneoplastic leukocytosis in cervical cancer (Table 2, page 12 of the revised version)
Also, I suggest making a flowchart or illustration to make readers understand easily about molecules mentioned in manuscripts. And another table to define immunologic cells and their functions is needed.
Reply: I have added an illustration to improve understanding of the main molecules mentioned in the article (Figure 1, page 3 of the revised version). Moreover, I have added a table to define immunological cells and their functions (Table 1, page 5 of the revised version)
Other Comments:
Topic is original & relevant in the field. It addressed enough gap in manuscript.
Reply: Thank you for your positive comments.
Review article is different from case report. It’s a little confusing to explain in detail about specific case. I suggest author focus more on systemic articles classification and related table and illustration if possible. Appropriate references are mentioned.
Reply: As indicated by you, I have better described the structure and the whole picture of the manuscript at the end of the first paragraph, clarifying that it aims to review the literature, and in the context of literature, considering the rarity of this condition, it will report an emblematic case report. Moreover, I have implemented the Discussion by adding a table (Table 2) with a detailed description and classification of articles available in literature on PLR associated with cervical cancers.
Author’s conclusion is “treatment strategies for cancer with PLR are few and rarely reported, our study could significantly contribute to the literature by expanding our understanding of the topic and providing a basis for further research” is a little different in “. our study represents an emblematic example and unique attempt that tested a mechanism-based targeted approach in such complex condition. More studies should be conducted to explore the phenomena that induce chemoresistance, rather than phenomena related to inflammation, to understand whether the same mediators of leukocytosis are the inducers of the peculiar chemo- and radio-resistance of these tumors”.
Reply: I have modified the conclusion as suggested by you (lines 584-588 of the revised manuscript)
Author should explain objective of this “Review” article more clearly.
Reply: Thank you for your comment. I have explained the objective and the structure of the Review at the end of the first paragraph (lines 43-51 of the revised version).
Reviewer 2 Report
The topic of the article is with significant implications for medical practice. First, the paper addresses the potential association of leukocytosis with a poor prognosis in cases diagnosed with cervical cancer.
The topic is relevant and exciting to the field of the journal. Therefore, the article makes a significant contribution to the field. The text is clear and easy to read. The overall paper is organized and well written. The authors review the specialized literature related to the approached subject in the first part. Discussions Section then follows the presentation of a case that presents similar findings. The literature reviews are insightful and informative.
The presented aspects sufficiently support the conclusions.
I have only a few remarks to make:
The abstract must not have more than 200 words.
Persistent leukocytosis was defined three times. Please use commas in numbers of four digits or longer.
Author Response
Point-by-point reply to Reviewer 2
Comments and Suggestions for Authors
The topic of the article is with significant implications for medical practice. First, the paper addresses the potential association of leukocytosis with a poor prognosis in cases diagnosed with cervical cancer.
The topic is relevant and exciting to the field of the journal. Therefore, the article makes a significant contribution to the field. The text is clear and easy to read. The overall paper is organized and well written. The authors review the specialized literature related to the approached subject in the first part. Discussions Section then follows the presentation of a case that presents similar findings. The literature reviews are insightful and informative.
The presented aspects sufficiently support the conclusions.
Reply: I would thank the reviewer for his very positive comments.
I have only a few remarks to make:
The abstract must not have more than 200 words.
Reply: I have shortened the abstract to 218 words (from more than 400 words), I hope it is acceptable.
Persistent leukocytosis was defined three times.
Reply: I have reported the definition of persistent leukocytosis one time (line 227); similarly, I have defined paraneoplastic leukemoid reaction (PLR) only one time (line 231).
Please use commas in numbers of four digits or longer.
Reply: thank you. I have revised the number of four digits or longer and used commas.
Reviewer 3 Report
This is a review paper about the pathogenetic mechanisms of PLR and its prognostic role in cervical cancer. In addition, the authors reported an emblematic case of advanced squamous cervical cancer associated with PLR and chemotherapy-resistance. I would make the following minor comments regarding the paper:
1. This is a well-written review paper. However, the authors need to shorten the length of manuscript and to clarify what they want to describe.
2. On the line 324 in page 7, a central and peripheral vascularization score of 4. What does it mean? High vascularity? High vascular flow? Please clarify the description and show the value of resistance index.
3. On the line 336 in page 7, please correct ‘infiltrative’ to ‘invasive’.
4. On the line 337 in page 7, the clinical stage according to the FIGO was IVA. According to FIGO staging 2018, stage IV is defined as the carcinoma has extended beyond the true pelvis or has involved (biopsy proven) the mucosa of the bladder of rectum. Was the stage IVA correct?
5. For the case, what is the reason why neither radiotherapy nor CCRT was considered as a primary treatment?
Other comments:
Topic is relevant to the field and it does address a specific gap in the field. It has detail reviews and it is an interesting case. Conclusions are consistent with the evidence and questions posed. References are appropriate too.Author Response
Reviewer 3
Comments and Suggestions for Authors
This is a review paper about the pathogenetic mechanisms of PLR and its prognostic role in cervical cancer. In addition, the authors reported an emblematic case of advanced squamous cervical cancer associated with PLR and chemotherapy-resistance. I would make the following minor comments regarding the paper:
- This is a well-written review paper. However, the authors need to shorten the length of manuscript and to clarify what they want to describe.
Reply: Thank you for your positive comments. I have added a paragraph at the end of introduction to better clarify the aim and the whole picture of the study (see lines 43-50 of the revised version). Moreover, I have shortened of about 3 pages the length of the manuscript, in particular I have shortened the sections 2 and 3 of at least 500 words.
- On the line 324 in page 7, a central and peripheral vascularization score of 4. What does it mean? High vascularity? High vascular flow? Please clarify the description and show the value of resistance index.
Reply: Score 4 means high vascular flow. Resistance index was 0.3. I have revised the text as follows: “The lesion showed a central and peripheral vascularization color score of 4, indicating a high vascular flow; resistance index was 0.3.” (lines 326-327 of the revised manuscript)
- On the line 336 in page 7, please correct ‘infiltrative’ to ‘invasive’.
Reply: I have corrected “infiltrative” into “invasive” (see line 338 of the revised version)
- On the line 337 in page 7, the clinical stage according to the FIGO was IVA. According to FIGO staging 2018, stage IV is defined as the carcinoma has extended beyond the true pelvis or has involved (biopsy proven) the mucosa of the bladder of rectum. Was the stage IVA correct?
Reply: The stage IV A is correct. The patient underwent a cystoscopy that confirmed with a biopsy the bladder involvement (see line 339 of the revised version).
- For the case, what is the reason why neither radiotherapy nor CCRT was considered as a primary treatment?
Reply: The patient was followed initially at another Centre where she started immediately platinum-based chemotherapy for the poor general condition and the presence of severe pain and disease-related symptoms. I agree with you that she should have been proposed at that time for CCRT and radiotherapy. It should be considered that after the first cycle she needed inward recovery for worsening clinical condition with uncontrolled pain and fever as indicated in the text. When the patient went to our attention the disease progression and extent with the associated worsening clinical condition, the related symptoms and the evidence of chemo-resistance led us to perform radical surgery. I have revised the text to better explain the reasons that the physicians of another Centre reported for starting firstly chemotherapy (see lines 341-342 of the revised manuscript).
Other comments:
Topic is relevant to the field, and it does address a specific gap in the field. It has detail reviews, and it is an interesting case. Conclusions are consistent with the evidence and questions posed. References are appropriate too.
Reply: Thank you for your positive comments and appreciation of our work and its significance. I hope to have addressed adequately your comments and suggestions.
Round 2
Reviewer 1 Report
Well corrected
thanks